# Investigating the potential neurotoxic effects of cell-free factors by *Batrachochytrium dendrobatidis* on locomotion in *Xenopus laevis*

Anna Lennon[1,*], Laura K. Reinert[2], Louise A. Rollins-Smith[2,3], Will Best[1] and Chase Kinsey[1,‡]

## ABSTRACT

Chytridiomycosis is a contributor to amphibian population declines. Diseased amphibians show symptoms of lethargy and loss of righting reflexes, likely due to an ion imbalance across the skin. However, it is possible developing zoosporangia release toxins that affect neuromuscular activity. Using *Xenopus laevis* as a model, we hypothesized that locomotor performance would be affected by injection of *Bd* supernatant factors. *X. laevis* were injected and then filmed performing a swimming escape response with high-speed cameras at 4 h, 24 h, and 1-week post-injection. Average maximum swimming velocity and escape latency were digitized using high-speed video. Despite no difference in escape velocity, there was a significant difference in escape latency 24 h post injection at both concentrations tested, $10^6$ and $10^7$ cell equivalents, though only differences at $10^6$ cell equivalents/ml supernatant persisted 1 week post injection. Changes in specific locomotor function suggest that there may be neurotoxins present, though the potential neurotoxins may exhibit neural circuit specificity across escape behavior. This study provides a method to test more purified extracts to determine whether *Bd* produces neurotoxic factors that could enter the blood stream and alter locomotion during a natural skin infection.

KEY WORDS: Neurotoxicity, Locomotor performance, Chytrid, Amphibians, Infection

## INTRODUCTION

Throughout the world, emerging infectious diseases are an extreme threat to both small and large-scale ecosystems, risking biodiversity loss that has a cascade of negative consequences (Daszak et al., 1999; Scheele et al., 2019; Luedtke et al., 2023; reviewed in Fisher et al., 2021). Chytridiomycosis – a fungal disease discovered in 1998 (Berger et al., 1998), is a more recent threat to amphibians in environments such as rain forests (Scheele et al., 2019; Luedtke et al., 2023; Weldon et al., 2004; reviewed in Daszak et al., 2000; reviewed in Skerratt et al., 2007). *Batrachochytrium dendrobatidis* (*Bd*) (Longcore et al., 1999) is one of two pathogenic species of chytrid fungi that cause the disease chytridiomycosis (reviewed in

Skerratt et al., 2007; Scheele et al., 2019; Sewell et al., 2021; reviewed in Rollins-Smith et al., 2022). Transmission can occur by contact with another infected individual or exposure to the waterborne zoospores (reviewed in Van Rooij et al., 2015). *Batrachochytrium dendrobatidis* zoospores infect the epidermis, and when the infection is severe, electrolyte transport through the epidermis from the environment is impaired. Eventually, this lack of essential ions leads to cardiac arrest (Voyles et al., 2007, 2009; Marcum et al., 2010; Campbell et al., 2012). Additional symptoms include lethargy, weight loss, skin sloughing, and loss of righting response (Daszak et al., 1999; reviewed in Van Rooij et al., 2015). Given the symptoms of loss of neuromuscular control, early speculation about mechanisms for the fatal disease suggested the release of possible toxins (Berger et al., 1998, 2005; Daszak et al., 1999). For example, fungal infection from *Aspergillus* species can have neurotoxic effects that damage neurons and neuroglial cells (Speth et al., 2000; reviewed in Richard, 2020). Furthermore, infection from *Bd* has been linked to ataxia, or uncoordinated muscle control, which is typical of neurological disorders (Mutschmann, 2015).

Locomotion is crucial for the survival of amphibians (Garland and Losos, 1994; Wilson et al., 2000, 2002; Kinsey et al., 2023). The locomotor ability of an amphibian can be indirectly impacted by parasitic infections, leading to depression of musculoskeletal performance (Goater et al., 1993; Goodman and Johnson, 2011). While poor ion exchange might have an effect on locomotor performance, it is also possible that locomotor depression is caused by neurotoxic effects produced by cell-free soluble factors. Few studies to date have examined potential neurotoxins produced by *Bd* (Hicks, 2013), nor quantified the functional impacts of cell-free soluble factors on locomotion. Examination of the genome of *Bd* demonstrated the presence of all necessary genes required to produce an epipolythiodioxopiperazine (ETP)-type toxin in *Bd*, although the toxin was not detected in culture media (Hicks, 2013). The destruction of cellular cytoplasm in epidermal cells occurring somewhat distant from the pathogen foci in the skin suggested possible toxin release (Berger et al., 2005). Another study of crayfish exposed to water that previously held *Bd* cells resulted in gill recession and mortality (McMahon et al., 2013). Therefore, the current study was undertaken to examine possible deleterious effects of *Bd* soluble products on locomotion as a proxy for neurotoxicity.

The objective of this study was to develop and subsequently use a method to test for neurotoxic effects on locomotor performance using soluble factors released by growing *Bd* cultures. *Xenopus laevis* has historically been used as a general model system in physiological studies with well-mapped anatomical, locomotor, and neural systems (Cannatella and de Sá, 1993; Straka and Simmers, 2012) and the symptoms of *Bd* infection are well documented. However, no study to date has attempted to quantify the locomotor performance in infected frogs, nor address potential mechanisms leading to poor performance, which have broad survival implications.

[1]Belmont University, Biology Department, Nashville, TN 37212, USA. [2]Department of Pathology, Microbiology and Immunology, Vanderbilt University School of Medicine, Nashville, TN 37235, USA. [3]Department of Biological Sciences, Vanderbilt University, Nashville, TN 37235, USA.
*Present address: Department of Biology, Indiana University, Bloomington, IN, USA.

[‡]Author for correspondence (chase.kinsey@belmont.edu)

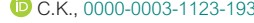 C.K., 0000-0003-1123-1931

To examine the effects of *Bd* on locomotion, we chose to use a *Bd* supernatant that includes extracellular products released by maturing sporangia that have been shown to induce apoptosis of lymphocytes (Fites et al., 2013). Previous literature suggested that *Bd* becomes lethal when infection burdens reach about 10,000 zoospore equivalents collected on a small swab (Vredenburg et al., 2010) with diseased frogs losing their righting reflex and experiencing physiological decline (Voyles et al., 2009; reviewed in Daszak et al., 2000 and Van Rooij et al., 2015). Because musculoskeletal physiology associated with self-righting is the same as that used for locomotion (i.e. same appendages and muscle group), we predicted that supernatant injections with greater cells/ml would result in a greater decrease of locomotor performance (i.e. swimming). By measuring locomotor performance in *Xenopus*, this study provides a method for testing neurotoxicity through a functional lens.

## RESULTS

The overall model for maximum velocity accounted for 25.5% of variance (marginal $R^2$=0.255) and 25.8% when including individual random effects (conditional $R^2$=0.258), indicating minimal among-individual variation. For escape latency, the model accounted for 21.1% (marginal $R^2$) and 27.7% (conditional $R^2$) of variance, with the 6.6% difference indicating moderate among-individual variation.

Repeatability analyses revealed minimal consistency in individual performance. Velocity showed effectively zero repeatability [unadjusted ICC=0.00, 95% CI: (0.00, 0.17); adjusted ICC=0.00, 95% CI: (0.00, 0.18)], with negligible treatment-specific values (control and $10^6$: 0.00; $10^7$: 0.06). Latency exhibited similarly low repeatability [unadjusted ICC=0.07, 95% CI: (0.00, 0.26); adjusted ICC=0.04, 95% CI: (0.00, 0.22)], with treatment-specific ICCs ranging from 0.00 to 0.11. These patterns indicate that variation in both metrics arose primarily from within-individual changes across time rather than stable individual differences.

Power analyses indicated that the design (16 frogs per treatment, three time points) provided 80% power to detect effects of Cohen's d≥0.58 (velocity: 0.18 m s$^{-1}$; latency: 0.16 s).

For velocity, observed treatment effects were small ($10^6$: d=−0.23; $10^7$: d=−0.36) and well below detection thresholds, with 95% CIs excluding differences >0.29 m s$^{-1}$, indicating adequate sensitivity to exclude moderate to large impairments. In contrast, latency showed large treatment effects that substantially exceeded detection thresholds (both treatments at 4 h: d=1.11), with the treatment×time interaction also characterized by large effect sizes (d=−0.66 to −1.38), confirming adequate power to detect the temporal pattern of impairment and recovery.

### Treatments effects

*Batrachochytrium dendrobatidis*-derived supernatant treatment had no significant impact on maximum velocity (F=0.737, P=0.485; Fig. 1). Frogs injected with sterile APBS exhibited comparable maximum velocities to those receiving $10^6$ cell-equivalent (difference=−0.070 m/s, SE=0.111, P=0.527) and $10^7$ cell-equivalent supernatant (difference=−0.109 m/s, SE=0.118, P=0.357). These findings suggest that *Bd*-derived supernatant, regardless of concentration, did not significantly impair locomotor performance during escape responses.

Linear mixed-effects model analysis of escape latency revealed a significant treatment×time point interaction (F=3.164, P=0.018; Fig. 2), indicating that treatment effects varied across the experimental time. Given this significant interaction, the main effect of treatment (F=5.137, P=0.011) cannot be interpreted independently and must be understood through time-point-specific comparisons.

Post-hoc analysis of simple effects revealed that treatment differences were most evident at early time points (Table 1, Fig. 3). At 4 h post injection, frogs receiving both $10^6$ cell-equivalent [difference=0.310 s, CI (0.060, 0.559), t=2.948, P=0.011, d=1.107] and $10^7$ cell-equivalent supernatant [difference=0.310 s, CI (0.044, 0.575), t=2.769, P=0.018, d=1.106] exhibited significantly longer escape latencies compared to controls, with no difference between the two treatment concentrations (P=1.000, d≤0.001). At 24 h, the $10^6$ cell-equivalent treatment group maintained significantly elevated latencies relative to controls [difference=0.314 s, CI (0.065, 0.563), t=2.989, P=0.010, d=1.122], while the $10^7$ cell-equivalent group no longer differed from controls [difference=0.126 s, CI (−0.140,

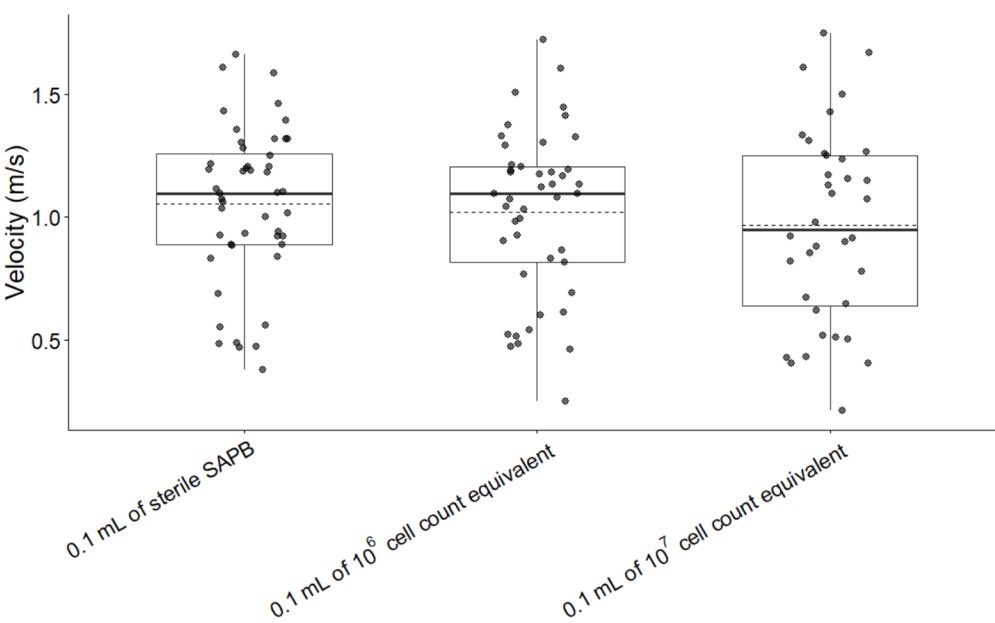

**Fig. 1. Boxplots of the effect of three treatments on frog swim velocity (m/s) for all frogs.** Boxes represent the first quartile, median, and third quartile. Solid line denotes median, dashed line denotes mean. A total of *n*=43 frogs were analyzed: 16 received the control (0.1 ml of sterile APBS), 15 received 0.1 ml of $10^6$ cell count equivalent supernatant, and 12 received 0.1 ml of $10^7$ cell count equivalent supernatant. There was no significant difference in average maximum velocity between $10^7$ cell count treatment (0.966 m/s±0.339) (*N*=12), $10^6$ cell count treatment (1.020 m/s±0.339) (*N*=15), and sterile APBS (1.050 m/s±0.316) (*N*=16).

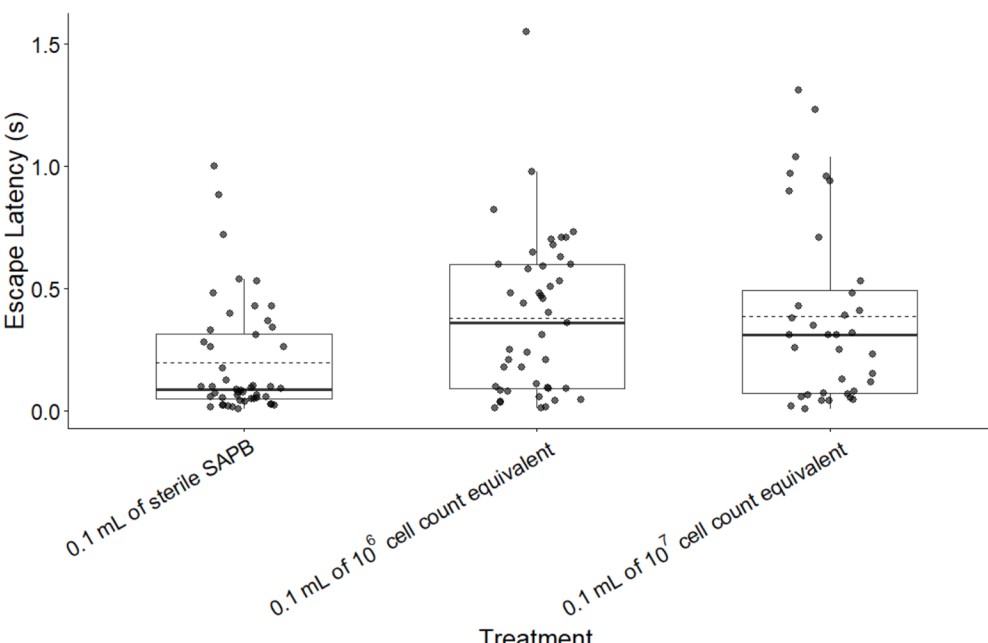

**Fig. 2. Boxplots of the effect of three treatments on frog escape latency (s) for all frogs.** Boxes represent the first quartile, median, and third quartile. Solid line denotes median, dashed line denotes mean. A total of $n=43$ frogs were analyzed: 16 received the control (0.1 ml of sterile APBS), 15 received 0.1 ml of $10^6$ cell count equivalent supernatant, and 12 received 0.1 ml of $10^7$ cell count equivalent supernatant. There was a significant difference in escape latency between $10^7$ cell count treatment (0.3878 s±0.374) ($N=12$), $10^6$ cell count treatment (0.381 s±0.332) ($N=15$), and sterile APBS (0.199 s±0.234) ($N=16$).

0.392), t=1.126, $P=0.500$, d=1.793]. By 1-week post injection, no significant treatment differences were observed (all $P>0.15$), suggesting recovery of normal escape response latency.

### Trial time point effects

To understand long term effects and potential recovery of supernatant injection in *X. laevis*, maximum velocity was calculated at 4 h, 24 h, and 1 week post injection. Trial time points significantly affected maximum velocity ($F=18.874$, $P<0.001$; Fig. 4), indicating a strong temporal influence on velocity. Post-hoc analysis revealed significant increases in velocity from the initial 4-h time point to both 24 h [difference=0.275 m/s, CI (0.114, 0.435), t=4.093, $P<0.001$, d=0.476; Table 2] and 1 week post injection [difference=0.402 m/s, CI (0.243, 0.562), t=6.013, $P<0.001$, d=0.891]. No significant difference was observed between 24-h and 1-week time points [difference= 0.128 m/s, CI (−0.032, 0.288), t=1.907, $P=0.143$, d=0.415]. These results indicate an initial recovery effect, with velocity increasing substantially after the first 4 h and stabilizing thereafter.

Trial time point significantly affected escape latency ($F=5.770$, $P=0.005$; Fig. 5), with latencies increasing from 4 h to 1-week

post injection (difference=0.363 s, SE=0.099, t=3.664, $P<0.001$). No significant difference was observed between 4-h and 24-h time points (difference=0.052 s, SE=0.099, t=0.522, $P=0.603$).

### Mass effects and interaction terms

Body mass was not a significant predictor of maximum velocity ($F=0.119$, $P=0.732$). No significant interaction was observed between treatment and trial time point ($F=0.994$, $P=0.416$; Table 1, Fig. 6), indicating that velocity increased similarly across all treatments over time. This uniform increase suggests that the observed improvements in velocity over time were independent of the *Bd* supernatant treatment.

Body mass was not a significant predictor of escape latency ($F=0.005$, $P=0.942$). As noted above, a significant interaction between treatment and trial time point was observed ($F=3.164$, $P=0.018$; Table 3, Fig. 3), with treatment effects diminishing over time, particularly for the $10^7$ cell-equivalent group.

### DISCUSSION

The goal of this study was to design a method for, and subsequently assess, the neurotoxic effects from *Bd* by conducting a behavioral assay on locomotor performance in the frog, *X. laevis*. Many of the symptoms associated with infection from *Bd* can be attributed to a decline in ion exchange across the epidermis (Voyles et al., 2009; reviewed in Van Rooij et al., 2015). However, loss of limb coordination and presumably decreased locomotor performance are not directly connected to loss of essential ions. Rather, we hypothesized that changes to muscular control may be driven by neurotoxins produced by *Bd*, as seen in other fungal infections (Speth et al., 2000; Mutschmann, 2015). Frogs were injected with a *Bd* supernatant and maximum swimming velocity was measured as a proxy for escape response across sterile (APBS) and two concentrations of supernatant and three time points: 4 h, 24 h, and 1-week post injection.

Differences in mass across specimens can invalidate results if not accounted for as mass and velocity are highly correlated (Garland, 1983). Our findings suggest that mass had a minimal influence on velocity under the conditions tested. This indicates that any velocity

**Table 1. Average maximum swimming velocity (m/s) and escape latency (s) with standard deviations for treatment groups across trial time points**

| Treatment | Trial time | Average maximum velocity (m/s) (Sd) | Escape latency (s) (Sd) |
|---|---|---|---|
| 0.1 ml of $10^7$ cell count equivalent | 4 h | 0.736 (0.343) | 0.370 (0.416) |
| | 24 h | 0.996 (0.361) | 0.238 (0.275) |
| | 1 week | 1.170 (0.397) | 0.556 (0.374) |
| 0.1 ml of $10^6$ cell count equivalent | 4 h | 0.777 (0.328) | 0.370 (0.419) |
| | 24 h | 1.160 (0.335) | 0.427 (0.308) |
| | 1 week | 1.120 (0.213) | 0.346 (0.229) |
| 0.1 ml of sterile SAPB | 4 h | 0.848 (0.257) | 0.061 (0.034) |
| | 24 h | 1.030 (0.305) | 0.113 (0.210) |
| | 1 week | 1.280 (0.229) | 0.424 (0.210) |

A total of $n=43$ frogs were analyzed: 16 received the control (0.1 ml of sterile APBS), 15 received 0.1 ml of $10^6$ cell count equivalent supernatant, and 12 received 0.1 ml of $10^7$ cell count equivalent supernatant.

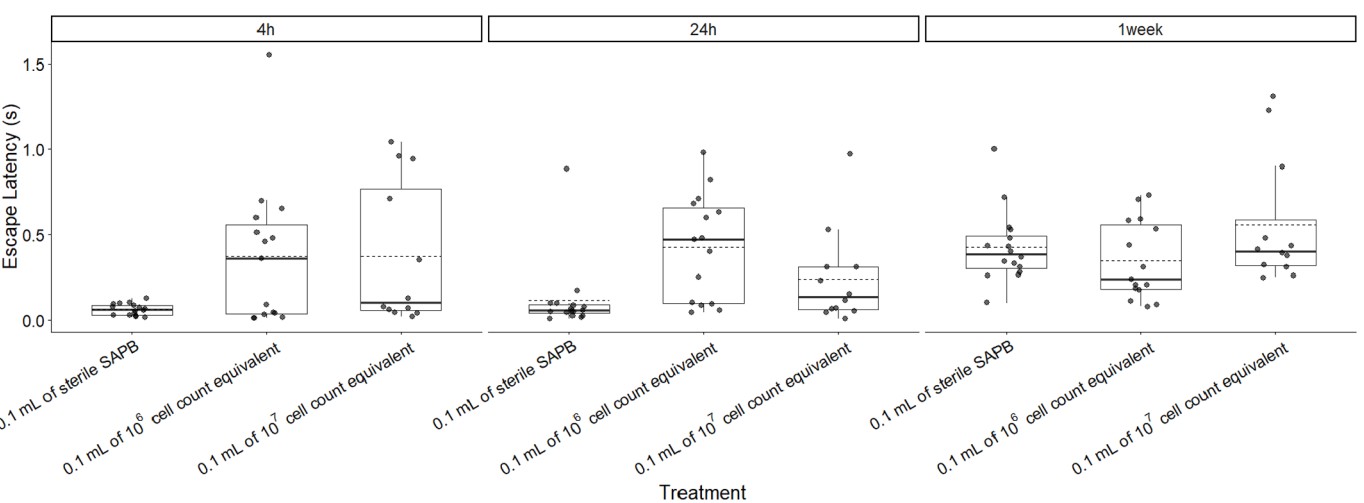

**Fig. 3. Boxplots of the effect of three treatments and three trial times on escape latency (s) for all frogs.** Boxes represent the first quartile, median, and third quartile. Solid line denotes median, dashed line denotes mean. A total of $n$=43 frogs were analyzed: 16 received the control (0.1 ml of sterile APBS), 15 received 0.1 ml of $10^6$ cell count equivalent supernatant, and 12 received 0.1 ml of $10^7$ cell count equivalent supernatant.

changes observed across time points and treatments were likely driven by other physiological or behavioral factors rather than body mass.

There were no significant differences in average maximum swimming velocity across the treatments; however, escape latency at $10^6$ cell-equivalent concentration was higher than $10^7$ cell-equivalent concentration or the control depending on time post-injection (Fig. 1). No difference in velocity between supernatant treatments and control APBS treatment could be driven by a lack of neurotoxic effect attributed to soluble products in the supernatant affecting fast twitch fibers, or it is possible the concentration of toxin from the *Bd* cells in this crude preparation was too low to produce measurable change. Despite no difference in escape velocity, there was a significant difference in escape latency 24-h post injection at both $10^6$ and $10^7$ cell-equivalent concentrations, though only differences at $10^6$ cell

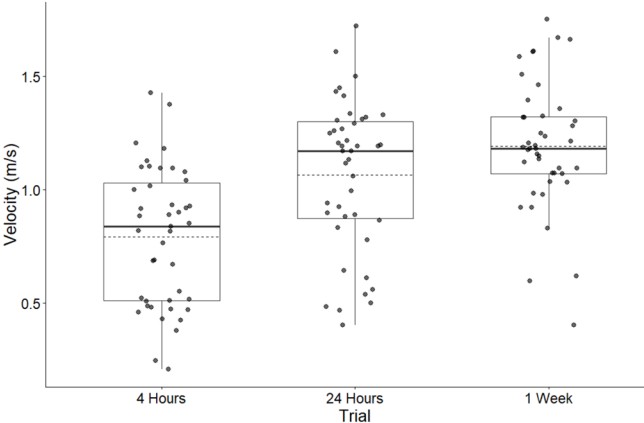

**Fig. 4. Boxplots of the effect of three trial times on frog swim velocity (m/s) of all frogs in all treatment groups ($N$=43).** Boxes represent the first quartile, median, and third quartile. Solid line denotes median, dashed line denotes mean. A total of $n$=43 frogs were analyzed: 16 received the control (0.1 ml of sterile APBS), 15 received 0.1 ml of $10^6$ cell count equivalent supernatant, and 12 received 0.1 ml of $10^7$ cell count equivalent supernatant. Average maximum velocity 4-h post infection (0.792 m/s±0.304) was significantly different from 24-h post injection (1.070 m/s±0.332) and 1-week post injection (1.190 m/s±0.284).

equivalent concentration persisted 1-week post injection. Previous studies have shown that neural control of variable escape responses is dependent on specific neural circuit morphology (reviewed in Domenici and Hale, 2019). Given the potential neurotoxic response, is possible the injected soluble products only affect a specific neural circuit associated with escape latency (i.e. latent period of muscle contraction), and not velocity (i.e. contractile speed).

Future studies should consider testing a more purified *Bd* supernatant preparation at higher concentrations. It is also possible that the fungus growing within the skin would produce and secrete toxins that would not be detected by our methods. That is, *Bd* growing in a rich tryptone broth culture medium does not produce the same array and quantities of substances that it does in pulverized frog skin (Rosenblum et al., 2012). Although *Bd* supernatants are known to contain several small metabolites that inhibit lymphocytes (Rollins-Smith et al., 2015; Rollins-Smith et al., 2019), no chemical, functional, or molecular characterization of the soluble products in the supernatant was performed in this study. Furthermore, the isolate used in these studies is the original type of isolate described in 1999 (JEL197) (Longcore et al., 1999) and has grown continuously since that time. Studies using other *Bd* isolates at varying concentrations and across muscle fiber types and muscle frogs should also be considered.

Despite no difference in average maximum swimming velocity between control and supernatant treatments, we observed that locomotor performance was reduced at 4-h post injection and showed marked improvement at 24 h and 1-week post injection. The sudden decrease at this early time point suggests that some

**Table 2. Pairwise post-hoc comparisons between trial time points for average maximum velocity with 95% confidence intervals and effect sizes**

| Comparison | Mean difference | 95% CI | $P$-value | Cohen's $d$ |
|---|---|---|---|---|
| 4 h versus 24 h | 0.275 | [0.114, 0.435] | **<0.001** | 0.476 |
| 4 h versus 1 week | 0.402 | [0.243, 0.562] | **<0.001** | 0.891 |
| 24 h versus 1 week | 0.128 | [−0.032, 0.288] | 0.143 | 0.415 |

A total of $n$=43 frogs were analyzed: 16 received the control (0.1 ml of sterile APBS), 15 received 0.1 ml of $10^6$ cell count equivalent supernatant, and 12 received 0.1 ml of $10^7$ cell count equivalent supernatant.

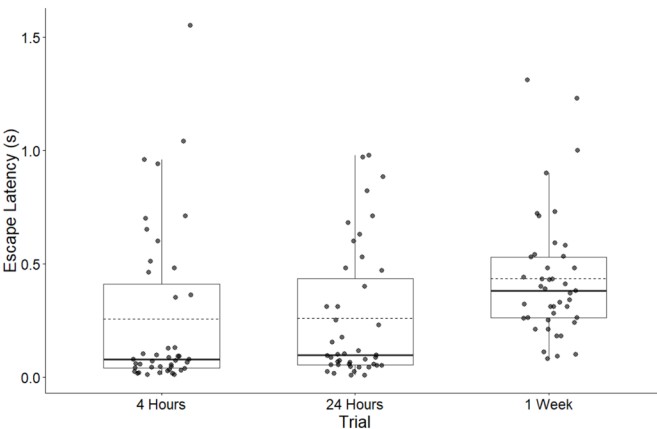

**Fig. 5. Boxplots of the effect of three trial times on frog escape latency (s) of all frogs.** Boxes represent the first quartile, median, and third quartile. Solid line denotes median, dashed line denotes mean. A total of $n=43$ frogs were analyzed: 16 received the control (0.1 ml of sterile APBS), 15 received 0.1 ml of $10^6$ cell count equivalent supernatant, and 12 received 0.1 ml of $10^7$ cell count equivalent supernatant. Escape latency 4-h post infection (0.225 s±0.357) and 24-h post injection (0.257 s±0.292) were significantly different from 1-week post injection (0.434 s±0.277).

other factor, such as injection stress, might have contributed to the changes. Acute stress, such as injection pain, is known to last for several hours post injection (Vodrážková et al., 2024). The injection could have created an acute stress response that influenced maximum velocity 4-h post injection and the frogs recovered within 24 h. Future work should consider adding an additional control with no injection to determine if any decrease in performance can be attributed to the injection. *Batrachochytrium dendrobatidis* infection is a chronic source of stress and extremely taxing on the skin integrity and skin function. This disease-related stress may also be associated with reduced locomotor performance over longer periods of exposure and may lower endurance. While we tested velocity and escape latency, future studies should consider evaluating endurance or locomotor metrics.

## Conclusion

Infection from *Bd* has led to global declines of amphibian populations. Previous studies clearly showed a link between poor ion exchange across the epidermis and symptoms including weight loss, skin sloughing, and cardiac arrest. Several symptoms, including ataxia and poor motor control are not directly explained by depression of epidermal ionic exchange and are typically associated with neurological infection. The purpose of this study was to develop a behavioral assay to detect changes in locomotor performance (i.e. motor control) using a supernatant from *Bd* as a source of possible neurotoxic molecules. Based on velocity and escape latency data across varying concentrations and time for *X. laevis*, we have evidence to suggest that poor motor control in diseased frogs is associated with soluble products produced by *Bd*, though we are unsure how or why the potential neurotoxin leads to differentiated degradation in specific escape responses. Future studies should continue to address the relationship between soluble factors, neurotoxicity, and locomotor depression in anurans.

## MATERIALS AND METHODS
### Supernatant preparation

A supernatant of the extracellular products of an actively growing *Bd* culture was prepared following published protocols (Fites et al., 2013). Briefly, *Bd* isolate JEL197 was expanded in 1% tryptone broth for about 10 days. Cells were pooled and mature cells were counted, centrifuged, and the pellet washed twice with sterile glass distilled water to remove remaining broth. The total population of mature cells (all cells beyond zoospore stage) was resuspended at a concentration of $10^7$/ml in sterile glass distilled water and incubated for 24 h at 21°C. After 24 h, the cells were centrifuged at 3000 RPM for 10 min and the supernatant collected. To ensure that no cells remained, the supernatant was filtered with a 0.2 μm pore-size filter. The filtered supernatant was frozen, lyophilized, and resuspended in one-tenth of the original volume of cells previously incubated in water. For injections into frogs to test for neurotoxin activity, the lyophilized supernatant was resuspended in amphibian phosphate buffered saline (APBS) (Ramsey et al., 2010). In previous studies, we observed that supernatants from approximately $10^7$ mature cells inhibited immune cell function *in vitro* (Fites et al., 2013). Thus, we chose to collect and test supernatants produced by about $10^7$ or $10^6$ mature cells for their effects on locomotion.

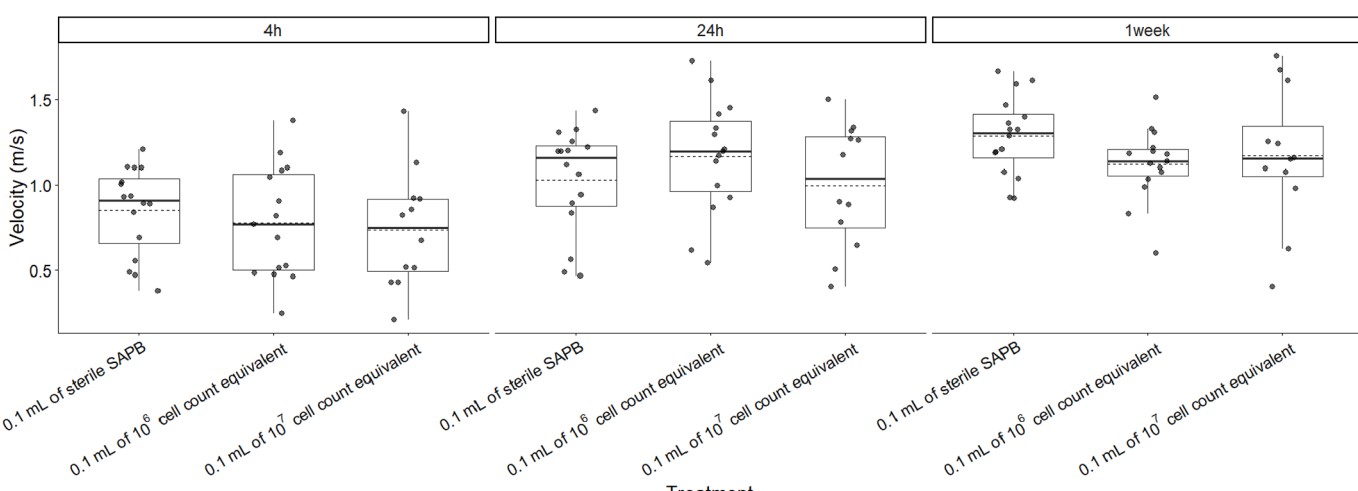

**Fig. 6. Boxplots of the effect of three treatments and three trial times on frog average maximum swimming velocity (m/s) for all frogs.** Boxes represent the first quartile, median, and third quartile. Solid line denotes median, dashed line denotes mean. A total of $n=43$ frogs were analyzed: 16 received the control (0.1 ml of sterile APBS), 15 received 0.1 ml of $10^6$ cell count equivalent supernatant, and 12 received 0.1 ml of $10^7$ cell count equivalent supernatant.

**Table 3. Pairwise post-hoc comparisons of escape latency between treatment groups at each time point with 95% confidence intervals and effect sizes**

| Time point | Comparison | Mean difference | 95% CI | P-value | Cohen's d |
|---|---|---|---|---|---|
| 4 h | Control versus $10^6$ | 0.310 | [0.060, 0.559] | **0.011** | 1.107 |
| 4 h | Control versus $10^7$ | 0.310 | [0.044 0.575] | **0.018** | 1.106 |
| 4 h | $10^6$ versus $10^7$ | <0.001 | [−0.269, 0.269] | 1.000 | <0.001 |
| 24 h | Control versus $10^6$ | 0.314 | [0.065, 0.563] | **0.010** | 1.122 |
| 24 h | Control versus $10^7$ | 0.126 | [−0.140, 0.392] | 0.500 | 1.793 |
| 24 h | $10^6$ versus $10^7$ | 0.188 | [−0.457, 0.081] | 0.226 | 0.671 |
| 1 week | Control versus $10^6$ | 0.077 | [−0.327, 0.173] | 0.745 | 0.275 |
| 1 week | Control versus $10^7$ | 0.133 | [−0.134, 0.400] | 0.465 | 1.027 |
| 1 week | $10^6$ versus $10^7$ | 0.210 | [−0.589, 0.479] | 0.157 | 0.751 |

A total of $n$=43 frogs were analyzed: 16 received the control (0.1 ml of sterile APBS), 15 received 0.1 ml of $10^6$ cell count equivalent supernatant, and 12 received 0.1 ml of $10^7$ cell count equivalent supernatant.

## Experimental design

Fifty juvenile (Nieuwkoop and Faber stage 66; average mass=1.02 g; mixed sex) (Nieuwkoop and Faber, 1994) *X. laevis* frogs were purchased from *Xenopus* Express (Brooksville, FL, USA) and kept in tanks with water temperature at 28°C and lights set to a 12:12 schedule for 3 days prior to testing to ensure acclimation to laboratory housing conditions. Frogs were fed three times a week following feeding protocols provided by *Xenopus* Express and fasted 24 h prior to experimental trials. Due to mortality upon arrival, only 43 frogs were used in the study. Frogs were randomly separated into three groups (Group A, *N*=12, Group B, *N*=15, and Group C, *N*=16) and weighed in grams by quickly patting each frog with wicking material and placing in a tared container. Once weighed, Group C (*N*=16) was injected with 0.1 ml of sterile APBS. Group B (*N*=15) was injected with 0.1 ml of 1X supernatant equivalent to the products of about $10^6$ cells/ml, and Group A (*N*=12) was injected with 0.1 ml of 10X supernatant solution that contained the products of about $10^7$ mature cells/ml. Subcutaneous injections were administered under the skin on the dorsal side. Injection in this manner was expected to result in significant uptake into the blood within 1 h (Howard et al., 2010). All specimens were left to rest after their injection for 4 h before swimming trials. Swimming escape trials were conducted in 29.8 cm by 15.24 cm by 20.57 cm clear tanks and filmed with a dorsally placed high-speed GO-Pro Hero black eleven camera at 240 frames per second (Fig. S1). Using a net, an individual frog was placed in the tank that contained around 5 cm of water and was allowed to adjust to the environment for 2 min. To measure maximum escape velocity, the frog was prompted to perform an escape response by using a blunt probe on the dorsum to simulate a predatory attack. This process was repeated three times for all frogs, with the most immediate and best response (i.e. no collision with tank walls) saved for further video analysis. Repeated stimulation was avoided beyond the three trials and any additional handling of the frog both during and between experiments was minimized to avoid habituation. The process of weighing and recording escape responses was repeated for all specimens 24 h and 1-week post injection. Given the relationship between mass and velocity (Clemente and Richards, 2013), it was essential to weigh before each swimming test to account for changes. Additionally, weighing provided a health metric as *Bd* infection can negatively influence mass (reviewed in Daszak et al., 2000; reviewed in Van Rooij et al., 2015). Following all trials, the frogs were euthanized with a buffered solution of MS-222 at a concentration of 6 g per l. Individuals were then preserved for future research. Data analysis was performed in MATLAB using DLTdataviewer7 code (Hedrick, 2008). Frog escape response was digitized using digital landmarks placed distally on the rostrum. Average maximum velocity was determined for each individual by averaging the top three velocities from the escape response. Escape latency was determined by subtracting the video frame where the frog began to accelerate from the frame where the frog first received a stimulus. All protocols and use of animals were approved by the Belmont University (AUP 2023-001).

## Statistical analysis

Differences in average maximum velocities and escape latencies between treatments, trial time points, mass, and their interactions were tested using linear mixed-effects models (LMMs) with individual frog identity as a random intercept to account for repeated measurements. Random slopes for time were not supported due to limited repeated observations per individual ($n$=3 per frog). Fixed effects included treatment (control, $10^6$ cell-equivalent, $10^7$ cell-equivalent), time point (4 h, 24 h, 1 week), centered mass, and the treatment×time point interaction. Models were fit using restricted maximum likelihood (REML), and assumptions were verified through visual inspection of residual plots. Three outliers were identified but retained as none were classified as extreme.

One frog was missing a latency measurement at the 4-h time point and this value was estimated by linear interpolation between observed time points. Sensitivity analyses confirmed that excluding this frog did not alter results. Post-hoc pairwise comparisons used estimated marginal means with Tukey's adjustment. Effect sizes were reported as marginal and conditional $R^2$ for model fit and Cohen's $d$ for pairwise comparisons, with 95% confidence intervals calculated for all estimated differences. Repeatability (intraclass correlation coefficient, ICC) was calculated overall and within treatment groups using 1000 bootstrap iterations. Power analyses assessed minimum detectable effect sizes at 80% power using simulation-based and analytical methods. Analyses were performed using R (version 4.4.2; R Core Team, 2024).

## Acknowledgements

We are thankful for the facilities provided by both Vanderbilt University Medical Center and Belmont University in culturing zoospores and housing specimens. We are also thankful for members of the Locomotor Performance Lab at Belmont for helping with animal care.

## Competing interests

The authors declare no competing or financial interests.

## Author contributions

Conceptualization: A.L., L.A.R.-S., C.K.; Data curation: W.B., C.K.; Formal analysis: A.L., W.B.; Investigation: A.L., C.K.; Methodology: A.L., L.A.R.-S., C.K.; Project administration: C.K.; Resources: L.K.R., L.A.R.-S.; Software: C.K.; Supervision: L.K.R., C.K.; Validation: L.A.R.-S.; Writing – original draft: A.L.; Writing – review & editing: A.L., L.K.R., L.A.R.-S., W.B., C.K.

## Funding

Research from the Rollins-Smith lab was funded by National Science Foundation (grants IOS 2147467 and BII 2120084). All other funding was provided internally by Belmont University. Open Access funding provided by Belmont University. Deposited in PMC for immediate release.

## Data and resource availability

10.6084/m9.figshare.30260242. All relevant data and details of resources can be found within the article and its supplementary information.

## Peer review history

The peer review history is available online at https://journals.biologists.com/bio/lookup/doi/10.1242/bio.062325.reviewer-comments.pdf

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
