## [Peer Review File · Biology Open]

Investigating the potential neurotoxic effects of cell-free factors by *Batrachochytrium dendrobatidis* on locomotion in *Xenopus laevis*

Anna Lennon, Laura K. Reinert, Louise A. Rollins-Smith, Will Best and Chase Kinsey
DOI: 10.1242/bio.062325

Editor: Kendra Greenlee

Review timeline

Original submission:	15 October 2025
Editorial decision:	22 October 2025
First revision received:	22 December 2025
Accepted:	03 January 2026

Original submission

First decision letter

MS ID#: bio. 062325

MS Title: Investigating the potential neurotoxic effects of cell-free factors by *Batrachochytrium dendrobatidis* on locomotion in *Xenopus laevis*

Authors: Anna Lennon; Laura K. Reinert; Louise A. Rollins-Smith; Will Best; Chase Kinsey

Dear Dr Kinsey,

I have now reached a decision on the above manuscript.

The reviewer reports are shown at the bottom of this email or can be accessed, together with a copy of this decision letter, by going to:

As you will see, the reviewers raised a number of substantial criticisms that prevent me from accepting the paper at this stage.

They suggest, however, that a revised version might prove acceptable, if you can address their concerns. If you think that you can deal satisfactorily with the criticisms on revision, I would be pleased to see a revised manuscript. In particular, I would encourage you to focus on improving the transparency of the methods, including the recommended changes to the statistical analysis, and scaling back conclusions in light of the limitations of interpreting the negative results for the experiments. I agree with reviewer 2 that it would strengthen the paper if you could include additional behavioral metrics that may be in the existing dataset. I also agree that providing some confidence that there was neurotoxin present in the supernatant, either through discussion of other studies that have used similar methods or by analysis of existing supernatant, would strengthen the conclusions.

Reviewer 1

Comments for the author

Batrachochytrium dendrobatidis (Bd) is a fungus known to cause severe amphibian declines worldwide. While Bd infections are typically associated with impaired electrolyte transport leading to physiological dysfunction, the possibility that secreted neurotoxins might contribute to symptoms such as reduced coordination has received little experimental attention. Here, Lennon et al. test this hypothesis by injecting *Xenopus laevis* frogs with Bd cell extracts - assumed to contain potential neurotoxins - and assessing escape responses at different timepoints post injection. The authors further present their experimental approach as a method to detect neurotoxic effects in amphibians.

Overall, this is a well-written and clearly structured manuscript addressing an intriguing question. The study is reasonably replicated (with sample sizes of N = 12-16) and the authors communicate their results transparently. However, I have several concerns that should be addressed to strengthen the manuscript.

1) First, the main objective of the study is not entirely clear. It remains somewhat ambiguous whether the primary aim was to test the neurotoxin hypothesis or to develop and validate a behavioral method for detecting neurotoxic effects. These two aims are related but not interchangeable, and clarifying this distinction would help sharpen the study's rationale and interpretation.

2) Second, several important methodological details are missing, making it difficult to fully evaluate - and eventually replicate - the work. More information is needed about animal handling, housing conditions, randomization procedures, and behavioral testing protocols. Moreover, some aspects of the behavioral design may introduce substantial confounding factors (e.g., pre-trial handling, repeated stimulation during trials), which should (ideally have been avoided and) be clarified or discussed.

3) Injection with Bd cell extracts did not affect the escape response of the frogs. Such "negative" results are surely valuable, but I would caution against drawing strong conclusions about the unlikelihood of neurotoxins being involved in Bd pathology, given alternative explanations for the lack of observed effects (e.g., exposure duration, concentration, or behavioral sensitivity of the assay).

In my opinion, this study presents an interesting contribution, but requires clearer framing and justification of its objectives, fuller methodological transparency, and a more nuanced discussion of the findings. Please find my detailed line-by-line comments below. Any negative tone that may appear between the lines is entirely unintentional - my comments are meant to be constructive and supportive of the authors' work.

Abstract

L28-30: I suggest also explicitly saying that Bd-treated frogs did not differ in their behavioral responses from control frogs (which now must be inferred from the next sentence).

L32-34: It may come as a confusing surprise to readers that there actually already is a commonly accepted explanation for decreased muscular coordination. Personally, I would acknowledge this information higher up already (in L20-22) and then present your toxins-hypothesis as a potential alternative explanation that you aim to test here (cf. my opening paragraph).

Summary statement

L41-42: In the absence of a response, can we really be sure that this method works?

Introduction

L44-55: This section mentions often (perhaps too often?) that infectious diseases result in large-scale mortality, threaten biodiversity loss etc. Paradoxically, this has a diluting effect, so I would suggest streamlining this part.

L61-63: This seems to be the major motivator of your study but, in my opinion, we're skimming over it too quickly here. Can this point be developed a bit further?

L64: I suggest saying that locomotion is crucial for the survival of amphibians, rather than describing it as a "tool".

L67-69: It would be interesting here to give more explanation about such indirect effects. I assume you are mostly referring to the electrolyte transport you mentioned earlier. If so, I suggest integrating these two sentences into your previous paragraph (and deleting them here) and zooming in on locomotion (e.g. L64-66) only afterwards. The narrative would then flow from "infections are bad for amphibians" -> "they mostly affect amphibians by affecting electrolyte transport or other indirect effects" -> "an alternative mechanism could be the release of neurotoxins" -> "this would lead to locomotor effects" etc. This would clarify things while also helping to tackle my earlier request to develop the major motivator of your study somewhat more.

L74-81: I apologize for my naivety, but this makes me wonder whether there are no analytical methods (e.g., LCMS) to directly detect and identify the toxin, rather than exposing (i.e., sacrificing) animals to try to figure this out indirectly. Could you add some more motivation for this choice?

L82-83: This links back to my previous question - could you add here 'why' we need this method?

L95-96: Considering the widespread use of clawed frogs as model organisms, and considering that behavioral change is already widely used in biomedical, pharmacological, and eco(toxico)logical research, was there still a need to develop this method? It seems unlikely that there are no prior studies and published protocols using locomotor performance in *Xenopus* to assess (neurotoxic) effects of exposure. Hence, it would be good to justify this objective further.

Methods

L99: This step is not really to "determine the presence" of a secreted neurotoxin, but rather to prepare supernatant that (hopefully) contains neurotoxins. Right?

L117-118: Was allocation to these three groups done randomly?

L119: How long were the frogs kept in the lab between arrival and manipulation? And under which conditions were they kept?

L128: Please provide more information on the exact testing conditions (water chemistry, water temperature, light exposure, feeding status etc.). The study doesn't seem replicable at this point because info like this is missing.

L131-133: I'm not sure what you mean. Was each individual stimulated repeatedly until it performed an acceptable response? Or...? If so, how many times were they stimulated per timepoint, and is there any way to take into account potential confounding effects of learning, fatigue, and/or overstimulation (within and across timepoints)?

L134-136: I understand that mass and velocity may be correlated, but why did you weigh animals before the swimming test rather than immediately after? It seems like weighing them before could have introduced a confounding stress factor during the behavioral trial.

L139: Do you mean video analysis?

L146: If I understand well, this should be "average maximum velocities" to be exact. Perhaps it's more accurate, therefore, to refer to them as "maximum velocities" rather than "average velocities"? (This comment also applies elsewhere.)

Results

L160: It would be better to first discuss the interaction terms rather than the main effects, because main effects cannot be trusted (or should not be interpreted) in case of significant interactions. After saying there were no significant interactions, you can proceed by interpreting the main effects.

Figure 1: repeated-measures ANOVA compares the means among the groups, so it would be better to also indicate the mean for each group (instead of only the quartiles and the median). As per journal instructions, please also include the raw data points in the figure (also in the other figures).

L171: (For future research, it could be interesting to also include a behavioral trial before the injections, to help figure out whether the injection itself could have any influence on the behavioral response.)

L173: Should this be Figure 2 rather than Figure 3?

L187-189: I agree, but I think this interpretation would go better in the Discussion section.

Discussion

L195-196: In my opinion, it is necessary, here and in the Introduction section, to be more clear about the actual goal of your study. Was it to figure out whether toxins could be involved in *B. dendrobatidis* infections, or was it to design a method for detecting neurotoxic effects? One does not exclude the other, but currently it is not clear to me what you were targeting.

L206: (here and elsewhere) it seems to me that "escape response" would be a more accurate term than "locomotor activity".

L207-210: I agree that these are possibilities, but they are the only possibilities. For example, we can't really exclude the possibility that this method doesn't work very well. Perhaps the weighing before the behavioral trial and its impact on the subsequent escape response overshadows any effect of treatment. Or maybe animals weren't exposed long enough, and any behavioral effect would only become visible after a longer period. And so on. (I acknowledge that you discuss alternative explanations further on, but it seems nevertheless good to rephrase the statement a bit here, to make it a bit more open-ended.)

L229: I agree that the injection itself may have influenced the response. But other effects could also be at play. Perhaps frogs learned that they should escape more quickly (because otherwise they "keep being bothered by that probe"). Or perhaps it's more of a developmental thing, and they manage to swim away more quickly over time because their muscles become more developed, or...

Conclusions

L244-248: Given that a lot of alternative explanations are possible (see up), I'd water down this conclusion. I agree that your results do not support the toxins-hypothesis, but to say that the hypothesis is unlikely is, based on these results and in my opinion, worded too strongly. Absence of evidence isn't evidence of absence. :-)

Reviewer 2

Comments for the author

This study addresses the interesting and underexplored hypothesis that soluble products from *Batrachochytrium dendrobatidis* (Bd) could influence neuromuscular performance in amphibians. The authors use *Xenopus laevis* as a model to test for neurotoxic effects of Bd supernatants via escape response performance. The question is relevant given uncertainty about the proximate mechanisms of chytridiomycosis symptoms beyond ion imbalance. The manuscript is clearly written, methodologically transparent, and provides a potentially useful assay framework for future work. However, there are a few issues in the experimental design, statistical analysis, and interpretation that could be improved:

Major Comments

1) The authors analyse their data using repeated-measures ANOVA/ANCOVA. However, a linear mixed-effects model (LME) with individual identity as a random effect would be a stronger approach as it would allow testing for random slopes and among vs within individual variation. Along the same lines, it would be useful to show measures of repeatability across the time points and whether this differed among the treatments.

2) The study uses maximum escape velocity as the sole performance metric. While this is an appropriate first test, it represents only one aspect of locomotor function, specifically, burst performance under a startle stimulus. Other forms of locomotion (e.g. routine swimming, endurance, or righting response latency) could be differently affected by potential neurotoxins, especially if different muscle fibres types are differentially affected by any toxic effects. Even within the current dataset, additional kinematic measures such as response latency would strengthen the interpretation.

3) No chemical or molecular characterization of the Bd supernatant was performed. Without confirming that potentially active compounds were present at biologically relevant concentrations, it is difficult to interpret the null effect. Even basic quantification of protein concentration or reference to prior data on active metabolite levels would help contextualize the results, if there are still samples available

4) The effective sample sizes (12-16 per treatment group) may be insufficient to detect subtle behavioural effects given the variability in swimming performance. The authors should report an estimate of statistical power or effect size sensitivity to clarify the confidence with which they can exclude moderate neurotoxic effects.

Minor Comments

- the boxplots are generally clear, but figure captions could include sample sizes. The font sizes on the axes are really small and should be increased.

- the background info in the intro is good but more rationale for using *Xenopus laevis* would be useful

Reviewer's Responses to Questions

Experimental quality

Does each figure have the proper controls?

If 'No', please indicate reasons in Comments for Author box below.

Reviewer #1:

- Yes

Reviewer #2:

- Yes

Were the data analyzed using appropriate statistical tests?

If 'No', please indicate reasons in Comments for Author box below.

Reviewer #1:

- Yes

Reviewer #2:

- No

Reproducibility

Were experiments performed using adequate number of biological replicates?
If 'No', please indicate reasons in Comments for Author box below.

Reviewer #1:

- Yes

Reviewer #2:

- Yes

Does the methods section provide sufficient detail to permit reproducibility?
If 'No', please indicate reasons in Comments for Author box below.

Reviewer #1:

- No

Reviewer #2:

- Yes

Completeness

Are the manuscript's conclusions supported by the data?
If 'No', please indicate reasons in Comments for Author box below.

Reviewer #1:

- No

Reviewer #2:

- Yes

Scholarship

Do the authors cite and discuss the merits of data that would argue for and against their conclusion?
If 'No', please indicate reasons in Comments for Author box below.

Reviewer #1:

- Yes

Reviewer #2:

- Yes

Does the manuscript title & abstract accurately reflect the contents of the manuscript, without hyperbole?
If 'No', please indicate reasons in Comments for Author box below.

Reviewer #1:

- Yes

Reviewer #2:

- Yes

First revision

Author response to reviewers' comments

Biology Open Reviewer Comments:

Reviewer 1: *Batrachochytrium dendrobatidis* (Bd) is a fungus known to cause severe amphibian declines worldwide. While Bd infections are typically associated with impaired electrolyte transport leading to physiological dysfunction, the possibility that secreted neurotoxins might contribute to symptoms such as reduced coordination has received little experimental attention. Here, Lennon et al. test this hypothesis by injecting *Xenopus laevis* frogs with Bd cell extracts - assumed to contain potential neurotoxins - and assessing escape responses at different timepoints post injection. The authors further present their experimental approach as a method to detect neurotoxic effects in amphibians.

Overall, this is a well-written and clearly structured manuscript addressing an intriguing question. The study is reasonably replicated (with sample sizes of $N = 12-16$) and the authors communicate their results transparently. However, I have several concerns that should be addressed to strengthen the manuscript.

1) First, the main objective of the study is not entirely clear. It remains somewhat ambiguous whether the primary aim was to test the neurotoxin hypothesis or to develop and validate a behavioral method for detecting neurotoxic effects. These two aims are related but not interchangeable, and clarifying this distinction would help sharpen the study's rationale and interpretation.

2) Second, several important methodological details are missing, making it difficult to fully evaluate - and eventually replicate - the work. More information is needed about animal handling, housing conditions, randomization procedures, and behavioral testing protocols. Moreover, some aspects of the behavioral design may introduce substantial confounding factors (e.g., pre-trial handling, repeated stimulation during trials), which should (ideally have been avoided and) be clarified or discussed.

3) Injection with Bd cell extracts did not affect the escape response of the frogs. Such "negative" results are surely valuable, but I would caution against drawing strong conclusions about the unlikelihood of neurotoxins being involved in Bd pathology, given alternative explanations for the lack of observed effects (e.g., exposure duration, concentration, or behavioral sensitivity of the assay).

In my opinion, this study presents an interesting contribution, but requires clearer framing and justification of its objectives, fuller methodological transparency, and a more nuanced discussion of the findings. Please find my detailed line-by-line comments below. Any negative tone that may appear between the lines is entirely unintentional - my comments are meant to be constructive and supportive of the authors' work.

Abstract

L28-30: I suggest also explicitly saying that Bd-treated frogs did not differ in their behavioral responses from control frogs (which now must be inferred from the next sentence). **Added.** **“Average maximum swimming velocity and escape latency were digitized using high speed video. Despite no difference in escape velocity, there was a significant difference in escape latency 24 hours post injection at both concentrations tested, 10^6 and 10^7 cell equivalents, though only differences at 10^6 cell equivalents/ml supernatant persisted one week post injection.”**

L32-34: It may come as a confusing surprise to readers that there actually already is a commonly accepted explanation for decreased muscular coordination. Personally, I would acknowledge this information higher up already (in L20-22) and then present your toxins-hypothesis as a potential alternative explanation that you aim to test here (cf. my opening paragraph).

We have added an additional statement as suggested. "...likely due to ion imbalance across the skin."

Summary statement

L41-42: In the absence of a response, can we really be sure that this method works? Previous studies show a marked decline in muscular control in frogs infected with *Bd*. Yet no study to date has individually addressed what potential aspect of infection drives degradation. It's possible a decline in locomotor performance is driven by infectious load or poor ionic exchange. It's also possible that the cellular products produced by *Bd* exert a neurotoxic effect. By isolating these soluble factors from the mature zoospore, we can determine if the products lead to locomotor degradation - suggesting some neuromuscular detriment - or if degradation is due to something else, to be tested in further studies (i.e. infectious load / ionic exchange). This study is the first step in determining the cause of locomotor decline in infected frogs. That said, we tried to update the language in the summary statement.

Introduction

L44-55: This section mentions often (perhaps too often?) that infectious diseases result in large-scale mortality, threaten biodiversity loss etc. Paradoxically, this has a diluting effect, so I would suggest streamlining this part. I have streamlined this first paragraph by removing redundant phrasing and combining several sentences.

L61-63: This seems to be the major motivator of your study but, in my opinion, we're skimming over it too quickly here. Can this point be developed a bit further? I pulled the lines about neurotoxicity and ataxia from the locomotion paragraph and added them here to provide additional context.

L64: I suggest saying that locomotion is crucial for the survival of amphibians, rather than describing it as a "tool". Text updated.

L67-69: It would be interesting here to give more explanation about such indirect effects. I assume you are mostly referring to the electrolyte transport you mentioned earlier. If so, I suggest integrating these two sentences into your previous paragraph (and deleting them here) and zooming in on locomotion (e.g. L64-66) only afterwards. The narrative would then flow from "infections are bad for amphibians" -> "they mostly affect amphibians by affecting electrolyte transport or other indirect effects" -> "an alternative mechanism could be the release of neurotoxins" -> "this would lead to locomotor effects" etc. This would clarify things while also helping to tackle my earlier request to develop the major motivator of your study somewhat more. Thanks for the insight. We have rearranged this portion of the introduction to better address your points above.

L74-81: I apologize for my naivety, but this makes me wonder whether there are no analytical methods (e.g., LCMS) to directly detect and identify the toxin, rather than exposing (i.e., sacrificing) animals to try to figure this out indirectly. Could you add some more motivation for this choice? I added additional motivation (ln 81-85). While direct analytical methods would certainly be useful, we were also interested in understanding the functional impact of the factors on performance. "...and the symptoms of *Bd* infection are well documented. However, no study to date has attempted to quantify the locomotor performance in infected frogs, nor address potential mechanisms leading to poor performance which have broad survival implications. "

L82-83: This links back to my previous question - could you add here 'why' we need this method?
 Additional reasoning added in ln 80-82

L95-96: Considering the widespread use of clawed frogs as model organisms, and considering that behavioral change is already widely used in biomedical, pharmacological, and eco(toxico)logical research, was there still a need to develop this method? It seems unlikely that there are no prior studies and published protocols using locomotor performance in *Xenopus* to assess (neurotoxic) effects of exposure. Hence, it would be good to justify this objective further. We attempted to be more explicit in why this study is necessary (ln 80-82 in the final version).

Methods

L99: This step is not really to "determine the presence" of a secreted neurotoxin, but rather to prepare supernatant that (hopefully) contains neurotoxins. Right? That's correct. I've updated to language to read: "The objective of this study was to develop and subsequently use a method to test for neurotoxic effects on locomotor performance using soluble factors released by growing *Bd* cultures (see methods)."

L117-118: Was allocation to these three groups done randomly? Yes, I've updated the text to reflect as follows. "Frogs were randomly separated into three groups (Group A, N = 12, Group B, N = 15, and Group C, N=16) and weighed in grams by quickly patting each frog with wicking material and placing in a tared container."

L119: How long were the frogs kept in the lab between arrival and manipulation? And under which conditions were they kept? I've added information related to this question and the request below. "Fifty juvenile (Nieuwkoop and Faber stage 66; Average mass = 1.02g; mixed sex) (Nieuwkoop and Faber, 1956) *Xenopus laevis* frogs were purchased from Xenopus Express (Brooksville, FL) and kept in tanks with water temperature at 28°C and lights set to a 12:12 schedule for three days prior to testing to ensure acclimation to laboratory housing conditions."

L128: Please provide more information on the exact testing conditions (water chemistry, water temperature, light exposure, feeding status etc.). The study doesn't seem replicable at this point because info like this is missing. Thanks for pointing out the missing information. I have added this information (ln 113-116)

L131-133: I'm not sure what you mean. Was each individual stimulated repeatedly until it performed an acceptable response? Or...? If so, how many times were they stimulated per timepoint, and is there any way to take into account potential confounding effects of learning, fatigue, and/or overstimulation (within and across timepoints)? I have added the number of stimulations used during the trial for each frog. It's possible (though highly unlikely) that habituation would have occurred across trials. If so, we would have likely seen a decrease in escape response performance as frogs would no longer be threatened by us. However, we saw the opposite occur, where velocity increased. Other longitudinal studies using similar methods also show a lack of habituation based on locomotor performance results.

L134-136: I understand that mass and velocity may be correlated, but why did you weigh animals before the swimming test rather than immediately after? It seems like weighing them before could have introduced a confounding stress factor during the behavioral trial. A major reason for weighing the frogs prior to any testing was to ensure we injected the frogs with a standardized amount of supernatant or APBS. The average mass was 1.02g and so we were able to determine that 0.1 mL of injected solution was appropriate. Yes, additional handling is likely to induce a stress response though this was unavoidable. We attempted to account for this by providing a four-hour resting period before beginning escape trials.

L139: Do you mean video analysis? I have added clarification.

L146: If I understand well, this should be "average maximum velocities" to be exact. Perhaps it's more accurate, therefore, to refer to them as "maximum velocities" rather than "average velocities"? (This comment also applies elsewhere.)

This has been updated throughout the manuscript

Results

L160: It would be better to first discuss the interaction terms rather than the main effects, because main effects cannot be trusted (or should not be interpreted) in case of significant interactions. After saying there were no significant interactions, you can proceed by interpreting the main effects. **Reordered to address interactions first.**

Figure 1: repeated-measures ANOVA compares the means among the groups, so it would be better to also indicate the mean for each group (instead of only the quartiles and the median). As per journal instructions, please also include the raw data points in the figure (also in the other figures). **Means added as dashed lines, raw data points also overlaid in each figure.**

L171: (For future research, it could be interesting to also include a behavioral trial before the injections, to help figure out whether the injection itself could have any influence on the behavioral response.)

I have added this idea at line 286-287: "Future work should consider adding an additional control with no injection to determine if any decrease in performance can be attributed to the injection."

L173: Should this be Figure 2 rather than Figure 3?

Thank you. This has been updated.

L187-189: I agree, but I think this interpretation would go better in the Discussion section.

I moved these sentences to the discussion and added a reference to explain why we included this statement.

Discussion

L195-196: In my opinion, it is necessary, here and in the Introduction section, to be more clear about the actual goal of your study. Was it to figure out whether toxins could be involved in *Bd* dendrobatidis infections, or was it to design a method for detecting neurotoxic effects? One does not exclude the other, but currently it is not clear to me what you were targeting.

We are targeting both! I've updated the language in the first sentence of the discussion to try and make this clearer. (ln 86 and 201): "The goal of this study was to design a method for, and subsequently assess, the neurotoxic effects from *Bd* by conducting a behavioral assay on locomotor performance in the frog, *Xenopus laevis*."

L206: (here and elsewhere) it seems to me that "escape response" would be a more accurate term than "locomotor activity".

Great point. I have replaced that term with escape response throughout.

L207-210: I agree that these are possibilities, but they are the only possibilities. For example, we can't really exclude the possibility that this method doesn't work very well. Perhaps the weighing before the behavioral trial and its impact on the subsequent escape response overshadows any

effect of treatment. Or maybe animals weren't exposed long enough, and any behavioral effect would only become visible after a longer period. And so on. (I acknowledge that you discuss alternative explanations further on, but it seems nevertheless good to rephrase the statement a bit here, to make it a bit more open-ended.)

I tweaked the sentence a bit to tone down some of the “this or that” language so that our ideas are merely possibilities and this language has been updated throughout. The escape performance methods are well established in the literature and our use of the control - which also underwent the same protocol sans supernatant - establishes a relative metric that we can use to make comparisons.

L229: I agree that the injection itself may have influenced the response. But other effects could also be at play. Perhaps frogs learned that they should escape more quickly (because otherwise they “keep being bothered by that probe”). Or perhaps it's more of a developmental thing, and they manage to swim away more quickly over time because their muscles become more developed, or...

I can certainly appreciate the alternatives you suggested here. I have added additional information in the methods to address the habituation of response, which we account for by minimizing unnecessary exposure to repeated stimuli. (ln 133-134). Muscle size would not have an effect across development because frogs were all from the same developmental stage. (see line 111).

Conclusions

L244-248: Given that a lot of alternative explanations are possible (see up), I'd water down this conclusion. I agree that your results do not support the toxins-hypothesis, but to say that the hypothesis is unlikely is, based on these results and in my opinion, worded too strongly. Absence of evidence isn't evidence of absence. :-)

We have toned down the language a bit. We also added an additional statement at the end of the conclusion to suggest that more work needs to be done here. : Future studies should continue to address the relationship between soluble factors, neurotoxicity, and locomotor depression in anurans.”

Reviewer 2: This study addresses the interesting and underexplored hypothesis that soluble products from *Batrachochytrium dendrobatidis* (Bd) could influence neuromuscular performance in amphibians. The authors use *Xenopus laevis* as a model to test for neurotoxic effects of Bd supernatants via escape response performance. The question is relevant given uncertainty about the proximate mechanisms of chytridiomycosis symptoms beyond ion imbalance. The manuscript is clearly written, methodologically transparent, and provides a potentially useful assay framework for future work. However, there are a few issues in the experimental design, statistical analysis, and interpretation that could be improved:

Major Comments

1) The authors analyse their data using repeated-measures ANOVA/ANCOVA. However, a linear mixed-effects model (LME) with individual identity as a random effect would be a stronger approach as it would allow testing for random slopes and among vs within individual variation. Along the same lines, it would be useful to show measures of repeatability across the time points and whether this differed among the treatments.

We initially attempted to fit models with random slopes for time point as suggested by the reviewer. However, given our sample size (n=43 frogs with 3 observations each), the random slope model was overparameterized and could not be estimated. We therefore used random intercept models, which appropriately account for the repeated-measures structure by allowing each individual frog to have its own baseline performance while estimating common fixed effects for treatment and time. This approach is consistent with recommendations for repeated-measures

designs where random slopes cannot be estimated. Measures of repeatability were added to the analysis as well.

2) The study uses maximum escape velocity as the sole performance metric. While this is an appropriate first test, it represents only one aspect of locomotor function, specifically, burst performance under a startle stimulus. Other forms of locomotion (e.g. routine swimming, endurance, or righting response latency) could be differently affected by potential neurotoxins, especially if different muscle fibres types are differentially affected by any toxic effects. Even within the current dataset, additional kinematic measures such as response latency would strengthen the interpretation.

Escape latency added to the analysis, using a linear mixed-effects model.

3) No chemical or molecular characterization of the Bd supernatant was performed. Without confirming that potentially active compounds were present at biologically relevant concentrations, it is difficult to interpret the null effect. Even basic quantification of protein concentration or reference to prior data on active metabolite levels would help contextualize the results, if there are still samples available. After collecting escape latency, we found a difference across concentration and time points. However, and to your point, we are unsure what specific compound within the supernatant is differentially affecting the frog's escape behavior. We point this out in the new discussion.

4) The effective sample sizes (12-16 per treatment group) may be insufficient to detect subtle behavioural effects given the variability in swimming performance. The authors should report an estimate of statistical power or effect size sensitivity to clarify the confidence with which they can exclude moderate neurotoxic effects.

Power analysis performed and added under results.

Minor Comments

- the boxplots are generally clear, but figure captions could include sample sizes. The font sizes on the axes are really small and should be increased. Figure captions changed to include sample size and font sizes increased.

- the background info in the intro is good but more rationale for using *Xenopus laevis* would be useful. Information has been added to the last paragraph of the introduction

Second decision letter

MS ID#: bio.062325R1

MS Title: Investigating the potential neurotoxic effects of cell-free factors by *Batrachochytrium dendrobatidis* on locomotion in *Xenopus laevis*

Authors: Anna Lennon; Laura K. Reinert; Louise A. Rollins-Smith; Will Best; Chase Kinsey

Dear Dr Kinsey,

I am happy to tell you that your manuscript has been accepted for publication in *Biology Open*, pending our standard publication integrity checks. It was accepted on 3rd January 2026.